# Observing System Experiments with an Arctic Mesoscale Numerical Weather Prediction Model

**Roger Randriamampianina [1],* [ID], Harald Schyberg [1] and Máté Mile [1,2]**

[1] Norwegian Meteorological Institute, Development Centre for Weather Forecasting, Oslo, P.O. Box 43, N-0313 Blindern, Norway; haralds@met.no (H.S.); matem@met.no (M.M.)

[2] Hungarian Meteorological Service, Unit of Methodology Development, P.O. Box 38, H-1525 Budapest, Hungary

* Correspondence: rogerr@met.no; Tel.: +47-91-60-43-59

**Abstract:** In the Arctic, weather forecasting is one element of risk mitigation, helping operators to have knowledge on weather-related risk in advance through forecasting capabilities at time ranges from a few hours to days ahead. The operational numerical weather prediction is an initial value problem where the forecast quality depends both on the quality of the forecast model itself and on the quality of the specified initial state. The initial states are regularly updated using environmental observations through data assimilation. This paper assesses the impact of observations, which are accessible through the global telecommunication and the EUMETCast dissemination systems on analyses and forecasts of an Arctic limited area AROME (Application of Research to Operations at Mesoscale) model (AROME-Arctic). An assessment through the computation of degrees of freedom for signals on the analysis, the utilization of an energy norm-based approach applied to the forecasts, verifications against observations, and a case study showed similar impacts of the studied observations on the AROME-Arctic analysis and forecast systems. The AROME-Arctic assimilation system showed a relatively high sensitivity to the humidity or humidity-sensitive observations. The more radiance data were assimilated, the lower was the estimated relative sensitivity of the assimilation system to different conventional observations. Data assimilation, at least for surface parameters, is needed to produce accurate forecasts from a few hours up to days ahead over the studied Arctic region. Upper-air conventional observations are not enough to improve the forecasting capability over the AROME-Arctic domain compared to those already produced by the ECMWF (European Centre for Medium-range Weather Forecast). Each added radiance data showed a relatively positive impact on the analyses and forecasts of the AROME-Arctic. The humidity-sensitive microwave (AMSU-B/MHS) radiances, assimilated together with the conventional observations and the Infrared Atmospheric Sounding Interferometer (IASI)-assimilated on top of conventional and microwave radiances produced enough accurate one-day-ahead forecasts of polar low.

**Keywords:** data assimilation; limited area model; Arctic observations; satellite observations

## 1. Introduction

Good knowledge of the Arctic environment is becoming more and more important due to the increasing activities such as ship traffic and resource exploitation in the region. These activities can be influenced by low temperatures, occurrences of high winds, fog, and darkness during the winter season. Marine operations might additionally be influenced by ocean waves, icing from sea spray, and the presence of sea-ice and icebergs. These environmental factors may occur in combination, thus increasing the operational challenges. In remote areas, the infrastructure and capability to manage difficult situations, hazards, or accidents may be distant or even unavailable in many cases.

This emphasizes a need for risk management for operators in the Arctic (see, for instance, Reference Lloyd's and Chatham House [1]). Weather forecasting is one element of risk mitigation, helping operators to have knowledge of weather-related risks in advance through forecasting capabilities at time ranges from a few hours to days ahead. Weather forecasts are input to ocean and sea ice forecasting as well, so they contribute to dealing with risks connected to ocean conditions. Forecasts from short-range Numerical Weather Prediction (NWP) models are the main tools in such service. Operational numerical weather forecasting is an initial value problem where the forecast quality depends both on the quality of the forecast model itself and on the quality of the specified initial state. The initial states are regularly updated using environmental observations through data assimilation (Lorenc [2], Daley [3]).

In the frame of the Arctic Climate Change, Economy and Society (ACCESS) project (Gascard et al. [4]), the Norwegian Meteorological Institute (MET Norway), among other scientific tasks, dealt with (1) describing the short-range monitoring and forecasting capabilities in the Arctic and (2) identifying the key factors limiting the monitoring and forecasting capabilities, and providing recommendations for key areas to improve the forecasting capabilities in the Arctic. These studies were conducted with the operational NWP model at MET Norway. The project period was interesting because MET Norway, like other HIRLAM (High-Resolution Limited Area Model) countries, changed its operational mesoscale HIRLAM model (Gustafsson [5]) to a high-resolution non-hydrostatic model based on the HARMONIE-AROME (H-A hereafter) (Bengtsson et al. [6]). For this reason the task 1) above was studied with both HIRLAM and H-A mesoscale models. This paper describes the use of the H-A model for parts of the task 1), while the description of the task 2), using also the H-A model and worked out through observing system simulation experiments, will be presented in a separate paper.

Many papers describe the impact of observations over the Arctic and high latitudes (Inoue et al. [7], Yamazaki et al. [8], Randriamampianina et al. [9]). Most of the published studies describe the usefulness of components of global observing system through campaign measurements (e.g., the Norwegian IPY-THORPEX (Kristjánsson et al. [10]) and ARCROSE (the Arctic Research Collaboration for Radiosonde Observing System Experiment (http://www.esrl.noaa.gov/psd/iasoa/node/123)). A sustained Arctic forecasting system for a service in support of Arctic operations and activities cannot rely on time-limited campaigns. It needs to rely on a sustained observing system. For the meteorological services, and in our particular case, such observations are presently distributed through the GTS (Global Telecommunication System, for conventional observations) and the EUMETCast (EUMETSAT's data dissemination system, for satellite observations).

The forecasting capabilities of a NWP model depends on several factors, such as the efficiency of the physical parameterization, the dynamics description, the data assimilation method, and the use of observations in the data assimilation. This paper studies the impact of the assimilation of conventional and satellite observations over the Arctic region using the H-A data assimilation (Gustafsson et al. [11]) and forecast systems (Bengtsson et al. [6]). During the ACCESS project, Norway and Sweden implemented and started to operate commonly a version of the H-A called MetCoOp model (Müller et al. [12]). Since the MetCoOp domain does not fully cover the Arctic (the area of interest for the ACCESS project), a new mesoscale model, called AROME-Arctic was implemented with a very similar configuration to the operational MetCoOp model. Among other settings, for example, the size of the two model domains is the same. This decision was taken in order to guarantee a smooth operational implementation and an easier service maintenance of the AROME-Arctic. See Section 2 for more details about the implemented model. The impact of different observation networks in the Arctic to the AROME-Arctic forecasting capabilities was studied through an observing system experiment (OSE).

Section 2 presents the data and methods, including the introduction of the NWP configuration, the availability of the observations over the region of interest, the processing of the observations, and the description of the performed experiments. Section 3 describes the obtained results, discussing through diagnostic, verification, and case study both the impact of observations on the AROME-Arctic

analyses, and the impact of observations on the forecast model. The last Section 4 summarises all findings and provides some discussion.

## 2. Data and Methods

### 2.1. The Assimilation and Forecasting System of AROME-Arctic

The study, described in Section 1, was suggested to be conducted with an operational NWP model at MET Norway. Although, we had a version of the operational HIRLAM mesoscale model at a 12-km horizontal resolution well covering the project's area of interest, this model system was in the process of being phased out. Thus, we found it meaningful to perform the study with a higher resolution, convection permitting model having the same configuration as our newly implemented, at that time, a high resolution NWP MetCoOp model. The new H-A based mesoscale model shares most of the configuration and settings with the MetCoOp model except that it is implemented over the Arctic as shown in Figure 1.

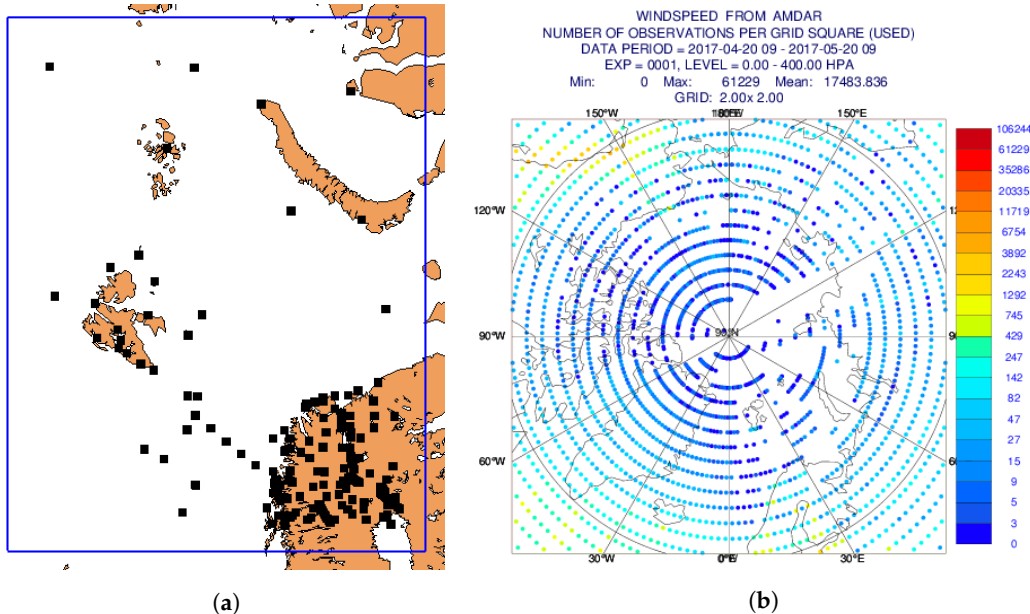

**Figure 1.** The usual coverage of conventional observations inside the AROME-Arctic domain (example taken for 4 August 2012, 12UTC) (**a**) and the number of observations available in ECMWF database (**b**).

It has a 2.5 km horizontal resolution (750 × 960 grid points) and 65 vertical levels from roughly 12 m (level 65) up to 10 hPa (level 1). The forecast model is based on H-A (Bengtsson et al. [6]) cy38h1.1 using a slightly modified version of the AROME physics developed at Météo-France (Seity et al. [13]) and non-hydrostatic dynamic (Bubnová et al. [14], Bénard et al. [15]). In this study, we use hourly ECMWF (European Centre for Medium-range Weather Forecasting) forecasts as lateral boundary conditions (LBC). Although, the default setting in the H-A and also in the MetCoOp system with an upper-air assimilation applies a spectral blending of the coupling (ECMWF) field at initial time (Dahlgren [16]) with the first-guess, this option is switched off in this study in order to better account for the impact of the observations. This means that we do not adjust the large-scale part of the initial state by bringing in the large-scale state described in the first LBC. Both the surface and the upper-air atmospheric fields are updated, respectively, using optimum interpolation for surface (Giard and Bazile [17]) and three-dimensional variational analysis (3D-Var) for upper-air (Fischer et al. [18], Gustafsson et al. [11]). Similarly to the operational MetCoOp data assimilation strategy, AROME-Arctic uses a 3-h intermittent cycle producing analyses at 00, 03, 06, 09 12, 15, 18, and 21 UTC. Surface (synop, drifting buoys, and ship), radiosonde, and aircraft observations were chosen from conventional observations. Due to a lack of conventional observations, satellite

observations are the only dominating ones for an NWP analysis in the Arctic. From satellite observations, the advanced microwave sounding unit-A (AMSU-A), unit-B (AMSU-B), the microwave humidity sounder (MHS), and the Infrared Atmospheric Sounding Interferometer (IASI) radiances were chosen. Further, we used the microwave data from NOAA (National Oceanic and Atmospheric Administration) and Metop satellites. Table 1 describes the use of different observations in the H-A data assimilation system.

$$J = J_b + J_o = \frac{1}{2}(x - x_b)^T B^{-1}(x - x_b) + \frac{1}{2}[y - H(x - x_b) - h(x_b)]^T R^{-1}[y - H(x - x_b) - h(x_b)] \quad (1)$$

The variational objective analysis consists of finding the solution $x$ at the minimum of the cost function $J$ in the Equation (1), where $x_b$ is the background or first-guess (a prior estimate of the state of the atmosphere, in our case, a 3-h forecast), $y$ is the vector of observations, $h$ is the fully nonlinear observation operator which projects the state of the atmosphere onto the space of the observations, $H$ is the tangent-linear version of the observation operator, and $B$ and $R$ are the covariance matrices of the background and observational errors, respectively. The background error statistics for the assimilation system were derived by downscaling global ensemble variational assimilation simulations, as described in Storto and Randriamampianina [19]. The computation of $B$ took into account the forecast performance over four seasons to avoid any discrepancies in the data assimilation accuracy in different seasons. The observation operator used for radiance assimilation is the RTTOV radiative transfer model (Saunders et al. [20]) in its version 10.2. For an IASI radiance simulation, the line-by-line transmittances were used (Matricardi [21]). Satellite radiances are bias-corrected in accordance with an adaptive variational scheme (Auligné et al. [22]) and with a specific daily coefficients aggregation and update as described in Randriamampianina et al. [9].

**Table 1.** The use of observations in the AROME-Arctic. Note, 10-m winds are assimilated over sea only.

| Type | Parameter (Channel) | Bias Correction | Thinning |
|------|---------------------|-----------------|----------|
| TEMP | U, V, T, Q | No | No |
| SYNOP | Z, V10m, U10m | No | Temporal and spatial |
| DRIBU | Z | No | Temporal and spatial |
| AIREP | U, V, T | No | 25 km horizontal |
| AMSU-A | Channels (see Table 2a) | Variational | 80 km horizontal |
| AMSU-B, MHS | Channels (see Table 2b) | Variational | 80 km horizontal |
| IASI | Channels (see Table 2c) | Variational | 80 km horizontal |

*2.2. The Availability of Observations over the Area of Interest*

The AROME-Arctic domain covers a poorly conventionally observed Arctic region. Figure 1b shows an usual aircraft AMDAR (Aircraft Meteorological DAta Relay) observation over the Arctic taken from the ECMWF observation monitoring system (https://www.ecmwf.int/en/forecasts/quality-our-forecasts/monitoring-observing-system#Conventional). Few ACARS (Aircraft Communications Addressing and Reporting System) observations can be also accessed, but these data are not used in this study. Figure 1a shows an usual coverage of conventional (all surface and upper-air) observations inside the AROME-Arctic domain. The situation is different when talking about satellite observations. Table 3, shows the availability of active, assimilated observations inside the domain for a randomly chosen day (7 December 2013).

*2.3. Processing of Satellite Radiances in AROME-Arctic Data Assimilation System*

All radiance observations are processed with full field of view (FOV) and all consecutive scan lines (Randriamampianina [23]). The active radiances are selected like other observations from moving platforms with a two steps thinning strategy. The first thinning allows radiances with a minimum distance by, respectively, 40 km, 60 km, and 60 km for AMSU-B/MHS, AMSU-A, and IASI.

The second thinning allows an average active radiance distance by 80 km. Table 2a–c shows the conditions for the assimilation of satellite radiances. The availability of different satellite paths inside the AROME-Arctic model domain is carefully monitored through a passive assimilation of the radiances. Then, all small paths with a non-satisfactory (non-converging to the nominal) bias correction after a month of monitoring are blacklisted for the given assimilation times. Hence, different satellites and channels are blacklisted differently at different assimilation times. Table 3 shows an example of active radiances during a day. For a correction of the radiance biases, the speed of adaptivity, the stiffness of the variational scheme, and the applied predictors compared to the default version (Auligné et al. [22]) are slightly modified according to Lindskog et al. [24]. The bias correction coefficients are aggregated and updated daily according to Randriamampianina et al. [9]. For cloudy IASI pixels, active channels having a peak above the cloud top were assimilated. The cloud detection used in H-A is a version of McNally and Watts [25] adapted to the IASI radiances as described in Collard and McNally [26].

*2.4. The Performed Experiments*

As explained above, this study started with the implementation of the H-A model in the Arctic. This means that, in order to have the best estimate of the model uncertainty—background error statistics B—needed for data assimilation, first, we tuned the forecast model to Arctic conditions. Here, we talk more about a technical rather than scientific adaptation of the model. Then, we prepared the data assimilation system with a careful estimation of the model background error. Normally, the observation and the background errors are evaluated through an iterative process using, for example, the method described in Desroziers et al. [27] and Storto and Randriamampianina [19] to estimate, respectively, the observation and the background errors. Due to the lack of time provided by the ACCESS project, this was not done. After checking the functionality of the variational scheme through a single observation experiment and diagnostic of the cost functions, we decided to keep the default observation errors.

**Table 2.** (a) The conditions for the assimilation of advanced microwave sounding unit-A (AMSU-A) channels in AROME-Arctic. Each condition is necessary but not sufficient. The obs-fg is the observation minus the simulated radiance in observation space; (b) the conditions for assimilation of advanced microwave sounding unit-B (AMSU-B)/humidity-sensitive microwave (MHS) channels in AROME-Arctic. Each condition is necessary but not sufficient. The obs-fg is the observation minus the simulated radiance in observation space; and (c) the conditions for the assimilation of Infrared Atmospheric Sounding Interferometer (IASI) channels in AROME-Arctic.

(a)

| Assimilation Conditions | AMSU-A Channels | | | | | | | | | | | | | | |
|---|---|---|---|---|---|---|---|---|---|---|---|---|---|---|---|
| | 1 | 2 | 3 | 4 | 5 | 6 | 7 | 8 | 9 | 10 | 11 | 12 | 13 | 14 | 15 |
| 3 < scan position < 28 | | | | | | x | x | x | x | x | | | | | |
| Over open sea | | | | | | x | x | x | x | x | | | | | |
| Over sea ice | | | | | | | x | x | x | x | | | | | |
| Over land | | | | | | | x | x | x | x | | | | | |
| Clear \|obs-fg\| ch 4 $\leq$ 0.7 K | | | | | | x | x | x | x | x | | | | | |
| Cloudy \|obs-fg\| ch 4 > 0.7 K | | | | | | | x | x | x | | | | | | |

(b)

| Assimilation Conditions | AMSU-B/MHS Channels | | | | |
|---|---|---|---|---|---|
| | 1 | 2 | 3 | 4 | 5 |
| 9 < scan position < 82 | | | x | x | x |
| Over open sea and \|obs-fg\| $\leq$ 5 K | | | x | x | x |
| Over land and \|obs-fg\| $\leq$ 5 K and model orography < 1000/1500 m for channels 3; 4 | | | x | x | |

|  |  |
| --- | --- |
| (c) | |
| **Assimilation Conditions** | **IASI Channels** |
| Over open sea | 38, 51, 63, 85, 104, 109, 167, 173, 180, 185, 193, 199, 205, 207, 212, 224, 230, 236, 239, 242, 243, 249, 252, 265, 275, 294, 296, 306, 333, 337, 345, 352, 386, 389, 432, 2919, 3008, 3014, 3069, 3087, 3098, 3207, 3228, 3281, 3309, 3322, 3339, 3438, 3442, 3484, 3491, 3499, 3506, 3575, 3582, 3658 |
| Over land | 38, 51, 63, 85, 104, 109, 167, 173, 180, 185, 193, 199, 205, 207, 212, 224, 230, 236, 239, 242, 243, 249, 252, 265, 275, 294, 296, 306, 345, 386, 389, 432, 2919, 3069, 3087, 3098, 3281, 3309, 3339, 3442, 3484, 3491, 3499, 3506, 3575, 3582, 3658, 4032 |
| Over sea ice | 51, 63, 85, 87, 104, 109, 167, 173, 180, 185, 193, 199, 205, 207, 212, 224, 239, 265, 275, 294, 306, 2701, 2819, 2910, 2991, 2993, 3002, 3008, 3014, 3027 |

**Table 3.** The number of active observations for data assimilation for 7 December 2013.

| | 00UTC | 03UTC | 06UTC | 09UTC | 12UTC | 15UTC | 18UTC | 21UTC |
| --- | --- | --- | --- | --- | --- | --- | --- | --- |
| Surf. Pressure, land | 25 | 18 | 34 | 19 | 31 | 19 | 32 | 17 |
| Surf. Pressure, auto | 60 | 60 | 51 | 60 | 55 | 59 | 53 | 60 |
| Surf. Pressure, ship | 2 | | 4 | | 3 | | 3 | |
| Surf. Wind, ship | 4 | | 8 | | 6 | | 10 | |
| AMDAR Temperature | 15 | 22 | 16 | 54 | 92 | 52 | 41 | 28 |
| AMDAR Wind | 64 | 42 | 32 | 108 | 184 | 102 | 80 | 56 |
| Dribu Pressure | 10 | 10 | 10 | 10 | 10 | 10 | 10 | 10 |
| Radiosonde Wind | 422 | | 138 | | 642 | | 144 | |
| Radiosonde Temperature | 132 | | 33 | | 185 | | 27 | |
| Radiosonde Humidity | 67 | | 20 | | 94 | | 18 | |
| METOP-A AMSU-A | | | 146 | 373 | 561 | 1104 | 1061 | 499 |
| METOP-A MHS | | | 142 | 1042 | 832 | 1015 | 809 | 377 |
| METOP-A IASI | | 15 | 6752 | 18,628 | 16,383 | 16,957 | 12,752 | 8157 |
| NOAA-15 AMSU-A | 9 | 346 | 465 | 349 | 802 | 174 | | |
| NOAA-18 AMSU-A | 187 | 630 | 395 | 316 | 302 | 608 | | |
| NOAA-18 MHS | 79 | 884 | 982 | 210 | 308 | 147 | | |
| NOAA-19 AMSU-A | 850 | 566 | 501 | 765 | 224 | | | 157 |

The study started with a downscaling of the ECMWF model to the AROME-Arctic resolution using dynamical adaptation without data assimilation. This run played as a reference experiment (ARCREF). This is in contrast to traditional data denial OSE studies, where the reference experiment usually is the one with the full set of observations. The implementation of the data assimilation system continued with adding more and more observations to the reference system as follows: ARCSURF—system with surface data assimilation only; ARCCONV—ARCSURF with added upper-air conventional observations assimilation; ARCAMSUA—ARCCONV with added AMSU-A radiances; ARCAMSUB—ARCCONV with added AMSU-B/MHS radiances; ARCATOVN—ARCCONV with added AMSU-A and AMSU-B/MHS radiances; and ARCIASI—ARCATOVN with added IASI radiances. Note that due to some unsatisfactory results, some of the experiments were restarted. Before the restart, a tuning of the data processing, for example, the blacklisting of more channels from

satellite instruments with a recomputation of the variational bias correction coefficients, was needed. For example, after discovering a problem related to the assimilation of AMSU-A radiances, we restarted all the AMSU-A related experiments before adding the IASI data into data assimilation. This also means that an implementation of the radiance observations into a regional data assimilation system requires thorough attention to different complex processes.

For verification purposes, long forecasts up to 48 h were conducted from 00 and 12 UTC runs. Twenty-five-day experiments were performed with four days of "warm-up" preceding each period. The impact of the observations on the analysis was estimated through a computation of the degrees of freedom for signal (DFS, Chapnik et al. [28]) as described in Randriamampianina et al. [9]. The impact of observations on analyses and forecasts was evaluated through the following techniques: (1) comparisons against observations and (2) a computation of the moist total energy norm (MTEN) loss caused by withdrawing different observations from the data assimilation system as described in Storto and Randriamampianina [29] and Randriamampianina et al. [9]. Furthermore, the results from the verification against conventional observations only (mainly surface and radiosondes) were validated using other verification technique, where much more observations were used. In fact, the developed technique is able to use all observations that can be processed by the H-A assimilation system. The verifying observations, including non-conventional ones, were taken from one the experiments to have comparable results.

## 3. Results

### 3.1. Impact of Observations on the Analysis System

The degrees of freedom for signal (DFS) are used to evaluate the sensitivity of the upper-air assimilation system (3-D-Var) to the different observations. The DFS is defined as the derivative of the analysis increments in observation space with respect to the observations used in the assimilation system (Chapnik et al. [28]). Details about our DFS diagnostic tool can be found in Randriamampianina et al. [9]. The absolute DFS values represent the information brought into the analyses by the different observation types in terms of the amount, distribution, instrumental accuracy, and observation operator definition. They offer an insight to the actual weight given to the observations within the assimilation system in terms of the self-sensitivity of the observations (i.e., sensitivity at observation location), but they do not provide any information on the spatial or cross- correlations between the observations and the analysis. The relative DFS (DFS normalized by the amount of the observations belonging to a specific subset) provides a theoretical value associated to each observation type, regardless of its actual amount and geographical coverage in the assimilation system. Usually, the DFS are computed with analyses well distant in time from each other to reduce the influence of interdependency between weather conditions prescribed in the model initial state. The following assimilation times and dates were chosen for DFS computation: 12 UTC for 6 December 2013 and 15 December 2013 and 00 UTC for 10 December 2013 and 19 December 2013. The final DFS values were calculated as the means over the four selected assimilation times. Analysing the results on the sensitivity of the assimilation system to different observation types (Figure 2), we can see the importance of the humidity and wind observations.

From conventional observations, only radiosondes provides humidity observation and from radiances, humidity sensitive channels are from AMSU-B/MHS and IASI. The sensitivity of the AROME assimilation systems to the humidity observations is also shown over the mid-latitude regions (e.g., Mile et al. [30]). The satellite data are the principal sources of observation over the Arctic. The more we use the satellite data the less is the relative influence of the conventional observation. Although, this is more visible only on surface observations, especially in case of drifting buoys (DRIBU), and less for the other observation types (Figure 2 bottom graph). Another interesting result is that the satellite radiances have a better influence on the analyses if used together, probably because of a better handling of the biases through the variational scheme.

*3.2. Impact of Observations on the AROME-Arctic Forecast Model*

In this section, we discuss the impact of different type of observations on the forecasts of the AROME-Arctic non-hydrostatic mesoscale model. On the one hand, analyses and forecasts are verified against observations. On the other hand, the sensitivity of the AROME-Arctic forecast model is estimated using the MTEN tool (Storto and Randriamampianina [29]).

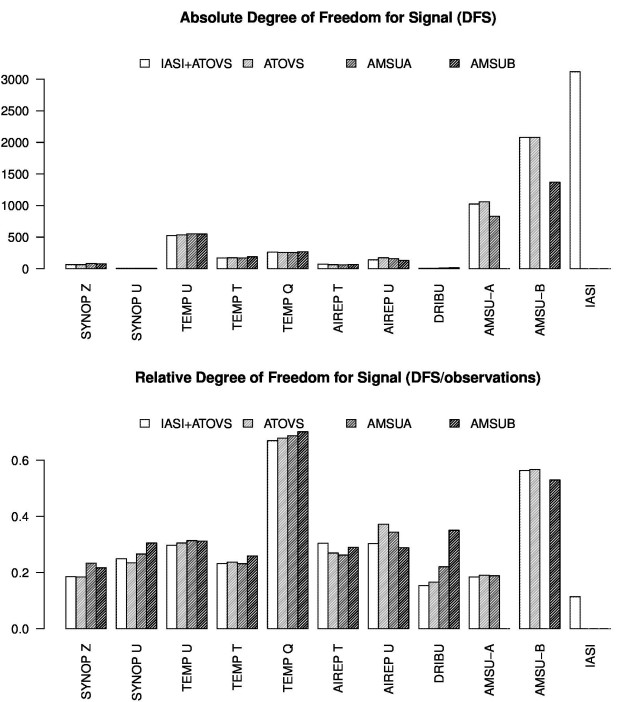

**Figure 2.** The absolute (top) and relative (bottom) degrees of freedom for signal (DFS) expressing the sensitivity of the assimilation system to different observed parameters in use, where IASI + ATOVS—conventional data + IASI + ATOVS; ATOVS—conventional data + AMSU-A + AMSU-B/MHS; AMSUA—conventional data + AMSU-A; and AMSUB—conventional data + AMSU-B/MHS.

3.2.1. Verification against Observations

Although, the best choice of verification method would suggest the use of independent observations (those not used in the assimilation process), over the studied region, we do not have enough observations to fulfill this requirement. Therefore, when analysing the verification results, we need to keep in mind that they are with respect to the used observations. This means that, at analysis time, the verification plots give an expression of how close the assimilation draws to the observations and not to the quality of the analysis. However, after some time into the forecast when the analysis increments have been propagated, the observations provide an independent verification. Figure 3 shows the usual verifying surface stations over the AROME-Arctic domain, where most of the stations are over land.

Still, in this study, we could draw a similar conclusion through different verification and diagnostic techniques. For example, there are usually 8 radiosondes present for verification of the upper-air forecast accuracy. The locations of these radiosondes seem to be very well-distributed inside the AROME-Arctic domain. To verify that, we worked out a verification technique where we take into account all active observations from the run with all observations, including non-conventional ones. The verification is done in the observation space. The verification against the observations shows mainly the following:

1.  Without data assimilation, the HARMONIE-AROME system provides less accurate forecasts, especially in the planetary boundary layer and lower troposphere.

2.  The impact of the surface assimilation is positive on the lower tropospheric temperature (Figure 4a) and geopotential (Figure 4b) in terms of the decrease of the root-mean-square error up to a 48-h forecast, and the impact is significant, respectively, up to 850 hPa and 700 hPa in the vertical. For a 10-m wind, the positive impact lasts up to a 48-h forecast. For all verified wind intensities, the impact is clearly positive (Figure 4c). For a 2-m temperature, the average decrease in error standard deviation (STDV) reaches in degrees Celsius from 1.5 at an initial forecast time to roughly 0.5 at a 12-h lead time (Figure 4d).

3.  Radiosonde observations alone or even with the aircraft data (relatively low amount) are not able to improve the forecast performance of the AROME-Arctic model compared to that of the pure downscaling of the ECMWF model (being hydrostatic but using a lot of observations) over the Arctic.

4.  Adding the radiances in the assimilation system clearly shows improvement (Figure 5). We show the performance on dew point temperature since it shows the impact of both the temperature and the humidity sensitive instruments (AMSU-A, AMSU-B/MHS, and IASI).

5.  The more observations we use in data assimilation system, the better is the accuracy of the forecasts (Figure 5a–c). Note that the vertical scores are the mean over different forecast ranges. See, for example, the different scores at 700 hPa. In performance quality, the best is the experiment with all observations (ARCIASI), then that with both microwave instruments (ARCATOVN), followed by the experiment with the humidity sensitive microwave (AMSU-B/MHS) (ARCAMSUB), and then the one with AMSU-A (ARCAMSUA). In this comparison, the lowest quality upper-air forecast is shown by running with conventional data only (ARCCONV). The difference in dew point temperature at 700 hPa between these runs is not significant.

6.  On Figures 5b and 6, one can see that a single satellite instrument (instrument groups) can provide a large enough impact, like the case of AMSU-B/MHS during this study period. In degree Celsius, more than 1 at analysis time and roughly 0.5 at 12- and 24-h forecasts of dew point temperature. A similar impact is also well-observed on the DFS studies. These results are also supported by the investigation related to the loss of total energy in the forecasts with respect to the withdrawn observations from the assimilation system (see Section 3.2.2 for more details).

7.  The verification against radiosonde observations is validated using the microwave humidity sensitive instrument (AMSU-B/MHS) data. Here, we compare the simulated and the observed brightness temperature in observation space. We can see a similar model performance in the lower troposphere through unbiased skill scores, in this case the root-mean-square error, used to compare radiosonde dew point temperature at 700 hPa and brightness temperature for channel 5 (183 ± 7 GHz) (Figure 7). The sensitivity (weighting function) of the AMSU-B/MHS channel 5 peaks around 700 hPa in the Arctic depending on the moisture content in the air. Since channel 5 data are assimilated only over sea, which is the dominating part of the model domain, we can say that the verification scores shown in the verification against radiosonde are representative.

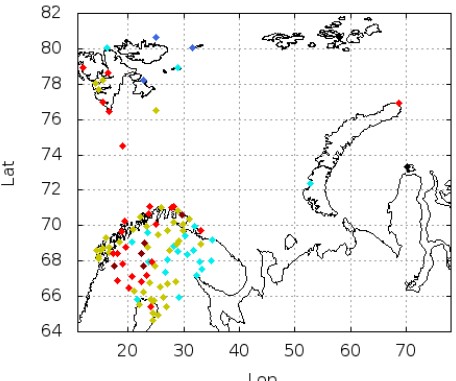

**Figure 3.** The coloured dots show the positions of the available verifying (surface and radiosonde) stations inside the AROME-Arctic model.

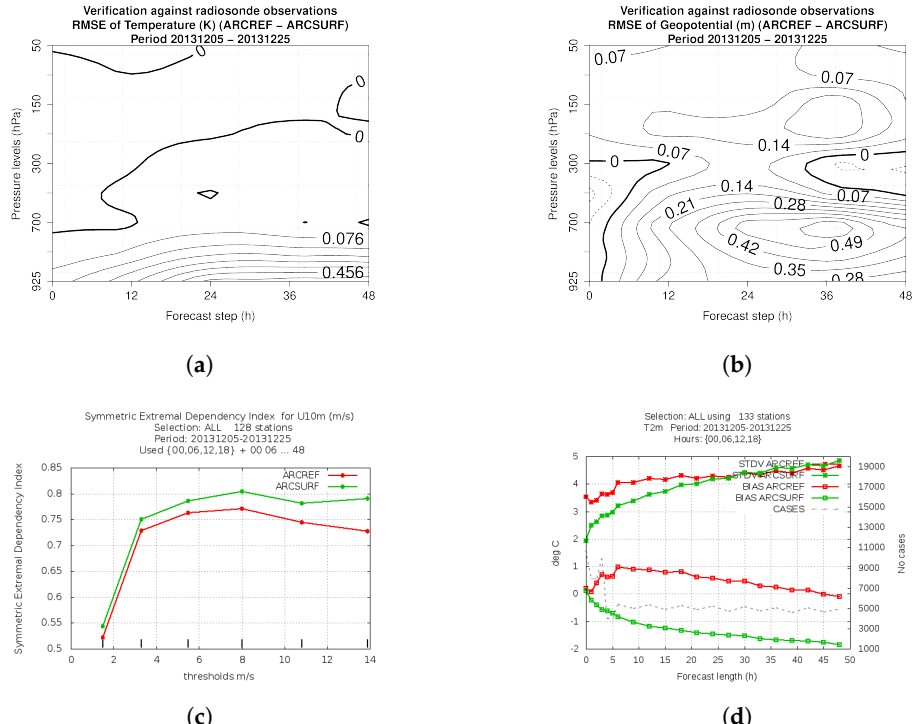

(**a**)                                                                     (**b**)

(**c**)                                                                     (**d**)

**Figure 4.** The impact of surface data assimilation on the temperature (**a**), geopotential (**b**), 10-m wind (**c**), and 2-m temperature (**d**). While for (**a**,**b**), positive/negative values shown a positive/negative impact, for (**c**), the higher the skill the better the impact and for (**c**,**d**) the green and red lines show, respectively, the experiment with and without surface data assimilation. RMSE, STDV, and BIAS, respectively, stand for root-mean-square error, error standard deviation, and bias.

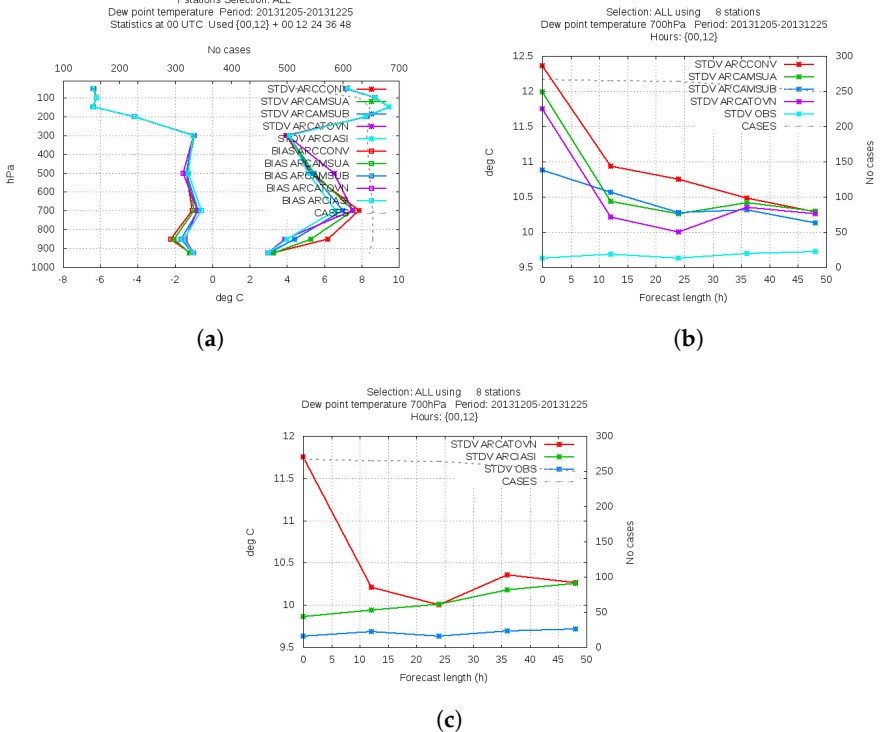

(**a**)                                                                     (**b**)

(**c**)

**Figure 5.** The verification of the analyses and forecasts of dew point temperature against observations expressed as mean scores over different model lead times (**a**) and score at the 700 hPa model level (**b**,**c**). Please refer to the legends for the different plots. STDV, BIAS, OBS, and CASES, respectively, stand for error standard deviation, bias, observation, and number of used cases in the verification.

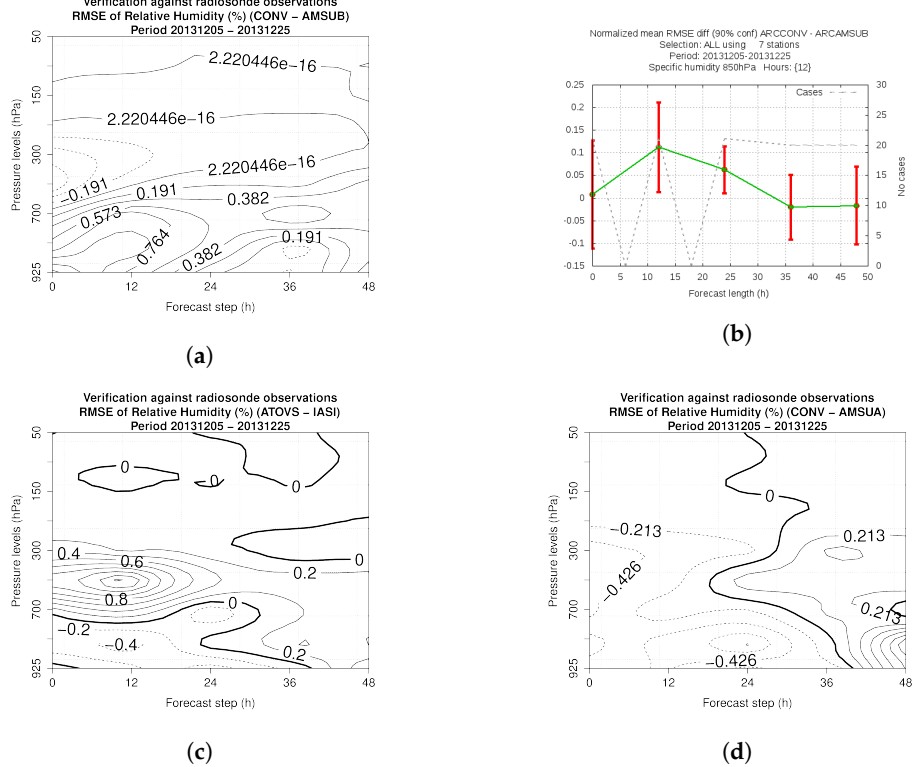

(**a**)

(**b**)

(**c**)

(**d**)

**Figure 6.** The relative root-mean-square error (RMSE) change in the forecast of relative humidity when adding AMSU-B/MHS (**a**), IASI (**c**), and AMSU-A (**d**) in the data assimilation system. The positive/negative values show positive/negative impacts of the satellite instruments. The graph in (**b**) shows the significant test applied to the normalized mean RMSE difference to the case at 850 hPa.

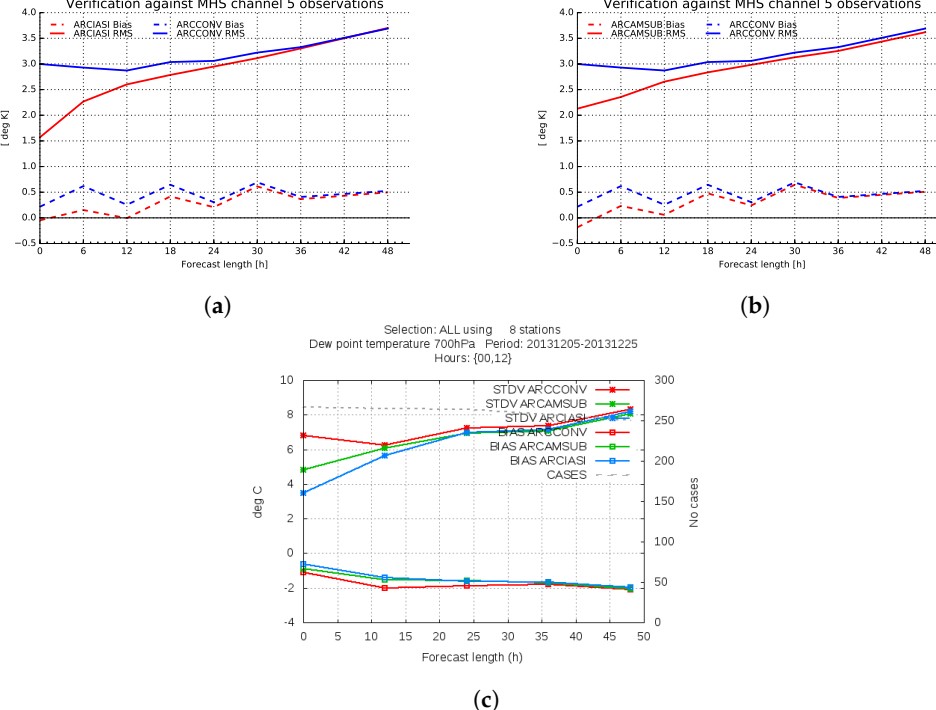

(**a**)

(**b**)

(**c**)

**Figure 7.** The verification against the AMSU-B/MHS channel 5 brightness temperature (**a**,**b**) and the verification against radiosonde observations (**c**). The horizontal axes in (**a**,**b**) show forecast lengths similar to the one in (**c**).

### 3.2.2. Sensitivity of the Forecast System to Different Observations

We use the technique developed by Storto and Randriamampianina [29] to assess the impact of different observation types on the forecasts. The sensitivity of the forecast model to the observations is defined by the change in a model space-based energy norm, between the experiment with all observations, and that where the evaluated observation set is taken out from the assimilation system. The impact of the observations is evaluated by means of a cost function, given as

$$J = \|M_t(x_{ctr}^a) - M_t(x_i^a), M_t(x_{ctr}^a) - M_t(x_i^a)\| \tag{2}$$

where $x_{ctr}^a$ and $x_i^a$ are the analysis from the "all-observation" experiment (called also control hereafter) and that with the withholding of the $i$th observing group, respectively, $M_t$ is the (fully nonlinear) forecast model operator, and $\|.\|$ stands for the moist total energy norm, defined as in Ehrendorfer et al. [31]:

$$\|x_t^i - x_t^{ctr}, x_t^i - x_t^{ctr}\| = \int_{\eta 0}^{\eta 1} \int_D (u^2 + v^2 + \frac{c_p}{T_r}T^2 + \frac{RT_r}{p_r^2}p^2 + \frac{L^2}{c_p T_r}q^2)\frac{\partial p_r}{\partial \eta}\delta\eta\delta D \tag{3}$$

where $u$, $v$, $T$, $p$, and $q$ are respectively the difference of the u- and v-components of wind, temperature, surface pressure, and specific humidity between the control forecast and the one without the ith set of observations; $c_p$, $R$, and $L$ are the specific heat at constant pressure, gas constant of dry air, and latent heat condensation; $T_r$ and $p_r$ are the reference temperature and reference pressure; and $\eta$ is the vertical coordinate. The norm is integrated over all the vertical levels between $\eta 0$ and $\eta 1$ and over the domain $D$, which may coincide with the whole model domain depending on the definition of the localisation operator $P$. In our case, for example, the AROME-Arctic domain was divided into four equal subdomains (see Figure 8).

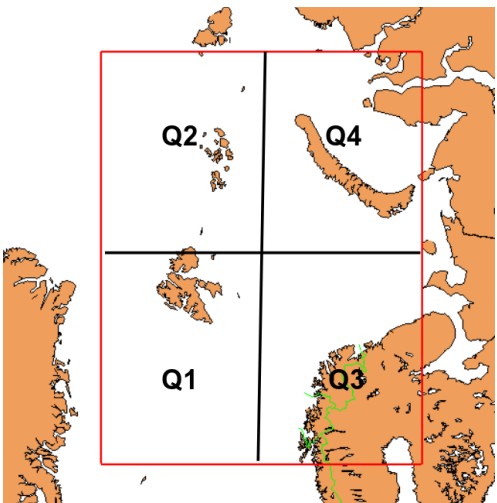

**Figure 8.** The AROME-Arctic domain and the subdomains for Moist Total Energy Norm computation.

This technique shows the quality loss associated with the withdrawn set of observations from the assimilation system. We applied this method to evaluate the impact of drifting buoys (DRIBU), aircraft (AIREP), radiosonde (TEMP), IASI (IASI), AMSU-A (AMSUA), and AMSUB/MHS (AMSUB) data. Dates distant enough apart were used to ensure ergodicity of the initial conditions, as recommended in Sadiki and Fischer [32]. The following dates and time were used: from 12 UTC for 6 December 2013 and 15 December 2013 and from 00 UTC for 10 December 2013 and 19 December 2013. The weather conditions along the forecasts at different chosen dates are shown in Figures 9–12.

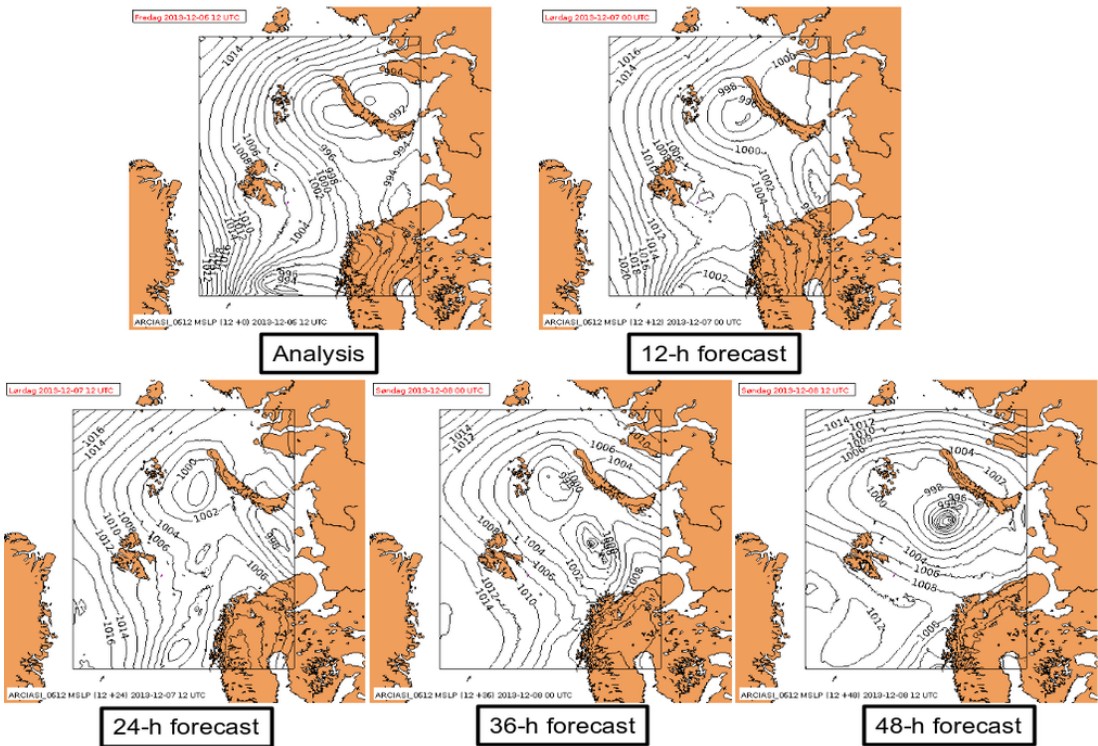

**Figure 9.** The analysis and forecast from 12 UTC, 6 December 2013. There is a polar low developing near Novaya Zemlya in these forecasts.

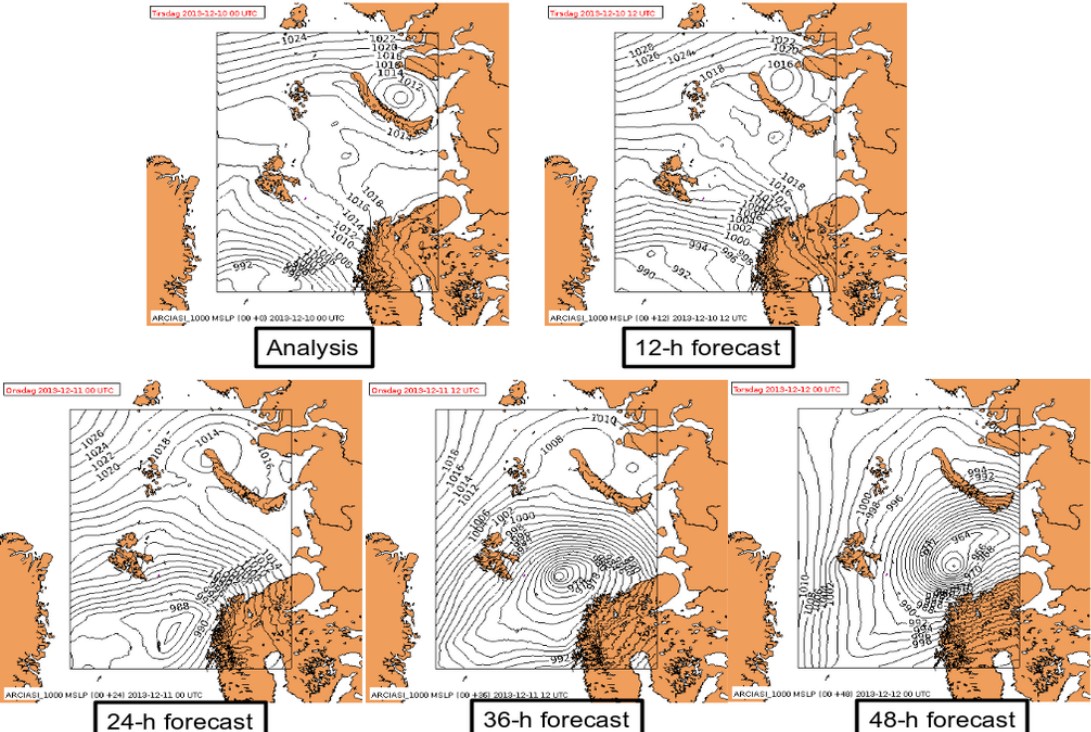

**Figure 10.** The analysis and forecast from 00 UTC, 10 December 2013. The dominating atmospheric systems are the decaying polar low near Novaya Zemlya and the developing synoptic scale cyclone moving relatively fast throughout the model domain.

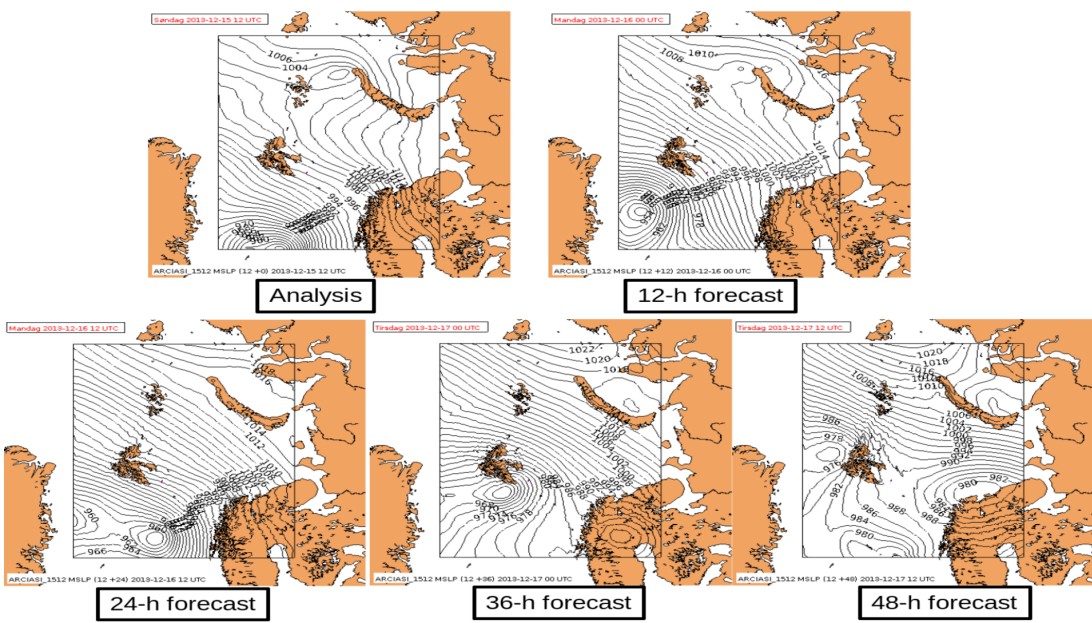

**Figure 11.** The analysis and forecast from 12 UTC, 15 December 2013. These forecasts are dominated mainly by large scale and stationary large gradient pressure system.

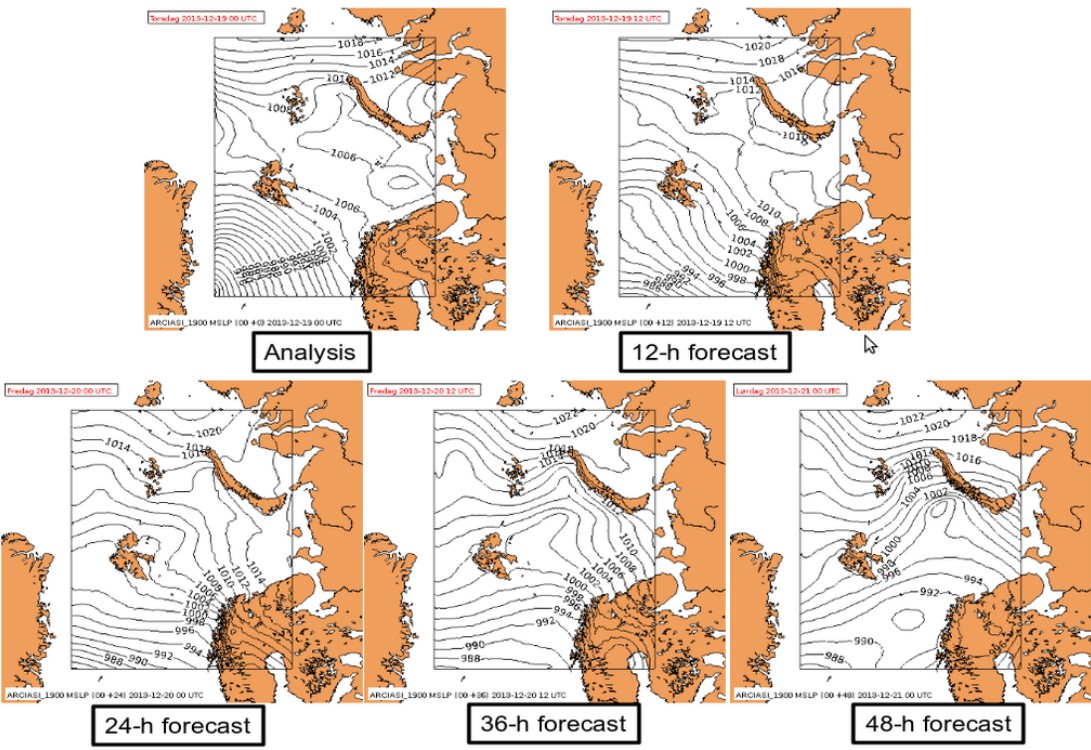

**Figure 12.** The analysis and forecast from 00 UTC, 19 December 2013. The forecasts show stationary atmospheric phenomena.

The sensitivity of the AROME-Arctic forecast to different observations depends on the dominating weather phenomenon over the whole domain (case of total norm). A separate computation of the moist total energy norms for different quarters of the domain shows that forecasting of severe weather systems (for example, a polar low—see the case of 6 December) is sensitive to all diagnosed observations. Note that the weather condition in first half of the study period was more influenced by different kinds of polar vortices than in the second one. This particular change of a dominating weather regime can be seen in the relative sensitivity of the forecast model in the two last verified dates (15 and

19 December). More likely, during these days, the model is driven by the lateral boundary condition, especially at a longer forecast range. The impact of different observation types along the forecast length was estimated. We highlight only few of them—6-h, 12-h, and 48-h forecasts (Figures 13–15, respectively). We can see that different types of observations play important roles in certain weather regime. Again, IASI and AMSU-B radiances seem to be the most influencing observations.

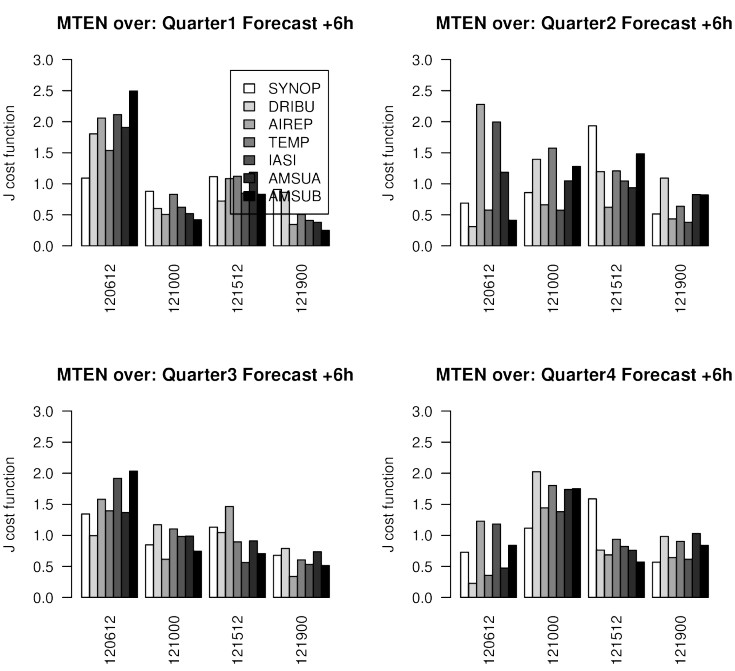

**Figure 13.** The sensitivity of the AROME-Arctic 6-h forecast to different observations for different dates and quarters of the model domain (see also Figures 8–12).

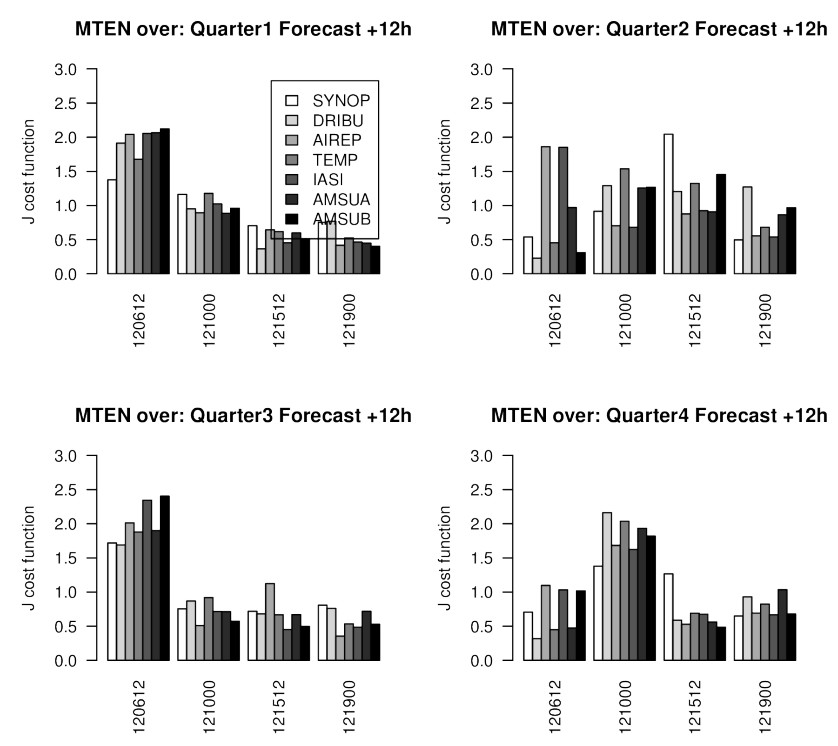

**Figure 14.** Same as Figure 13 but for a 12-h forecast (see also Figures 8–12).

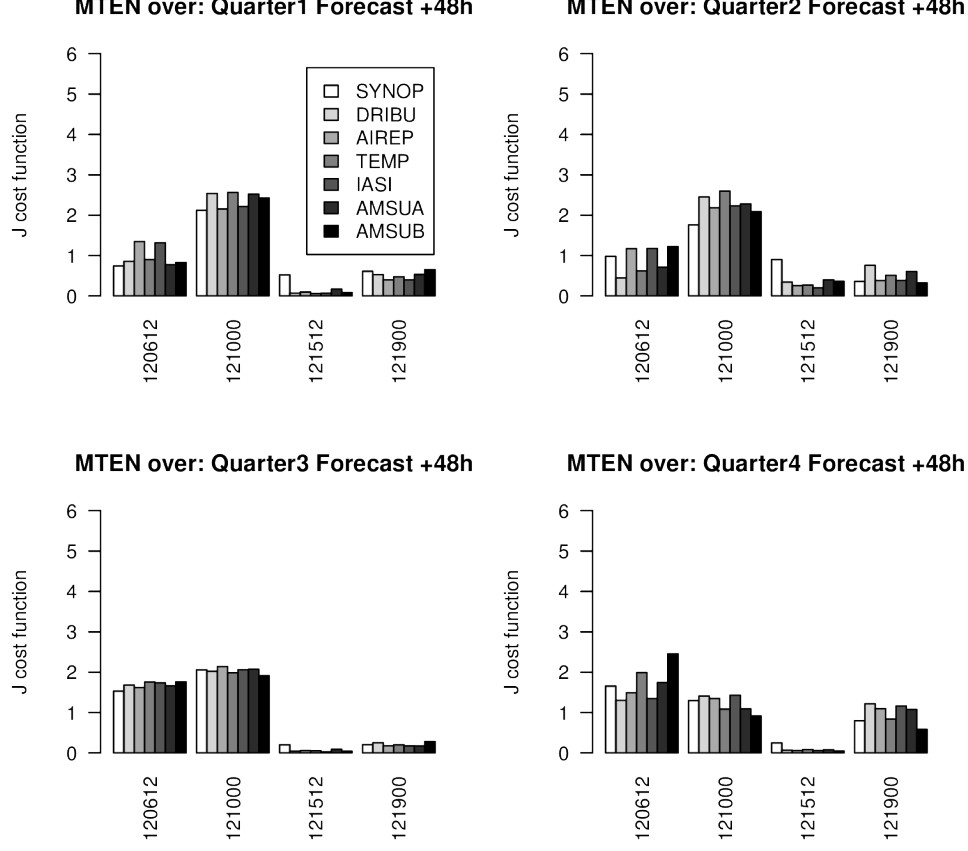

**Figure 15.** Same as Figure 13 but for a 48-h forecast (see also Figures 8–12).

### 3.3. Case Studies

Two cases of low-pressure systems were checked, where one of them—the case of 8 December 2013 at 12 UTC—was a polar low development. The second case—12 December 2013 at 00 UTC—was a fast-developing synoptic-scale cyclone passing through the AROME-Arctic domain within one and a half day time frame. Due to a lack of verifying datasets, the validation of the forecasting capability over the AROME-Arctic domain was very difficult. We did not have well-established gridded "observation fields" or radar data at the areas of interest, which, in both cases, are over seas. Taking into account this weakness in our study, we report only the first case, where we can base our evaluation to known theory. Nevertheless, in the second case, it could be seen that the precipitation forecast issued from the run with all observations included is very similar to the one issued from the run with conventional and AMSU-B/MHS observations.

The Polar Low Case—8 December 2013

Like in Randriamampianina et al. [9], among the different definitions of polar low, we took as reference the guidances, which suggest the existence of an upper-tropospheric synoptic-scale system together with the polar low developing at the lower tropospheric levels. Figure 16 illustrates the state of this polar low as forecasted two days earlier for 12 UTC 8 December 2013.

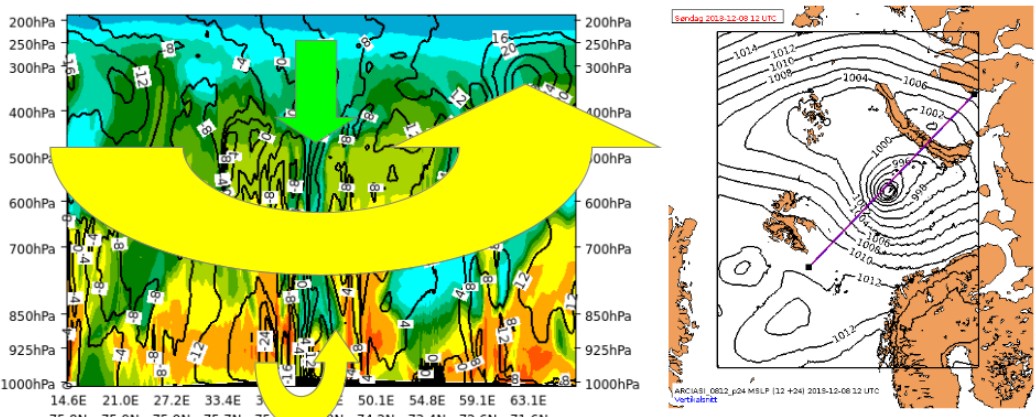

**Figure 16.** A vertical cross section of the relative humidity (coloured pattern) and a normal-wind field (black lines) along the line shown in the right hand side map. The larger circle shows the upper-tropospheric synoptic-scale cyclone, and the small one shows the polar low (acting below roughly 800 hPa). The cross section of humidity shows a dry air at the centre of the low as a signature of the stratospheric air intrusion, as found during the campaign observation (Linders and Sætra [33]; Kristjánsson et al. [10]. The plots are using a 24-h forecast, valid at 12 UTC 8 December 2013, from the run with all observations (ARCIASI).

In this case, we show the ability of different numerical solutions (with and without data assimilation) in forecasting the state of the polar low at 12 UTC on 8 December 24-h ahead (case chosen randomly). This polar low developed relatively slow and lasted relatively long to reach its mature stage (around 00 UTC 9 December) (see Figures 17 and 18), while it started to be visible in the analyses at 18 UTC on 6 December.

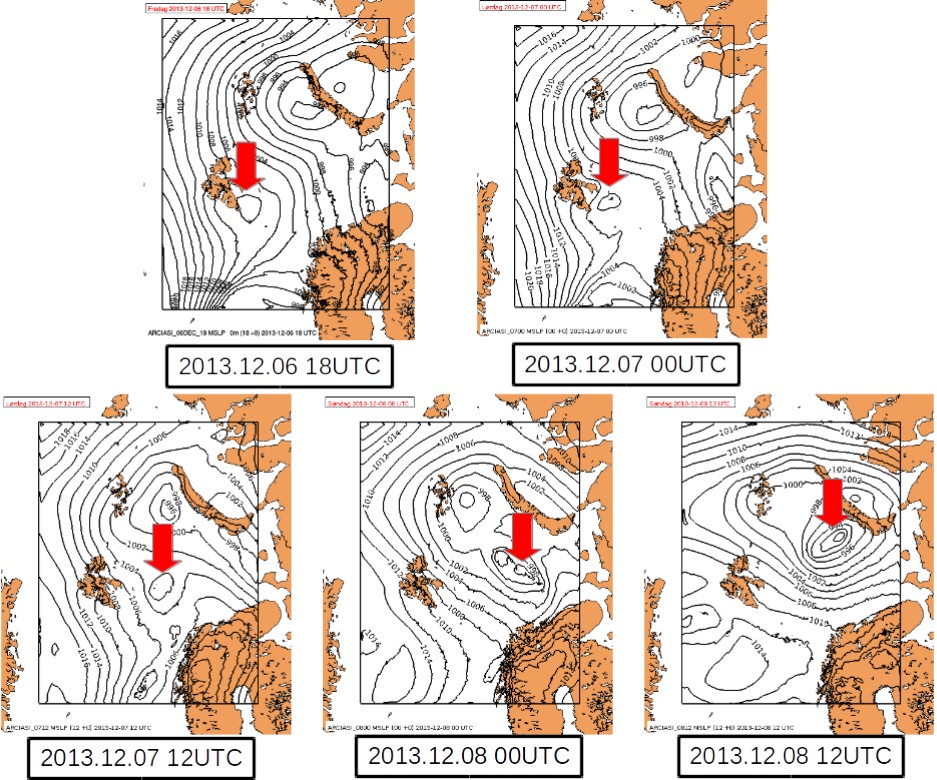

**Figure 17.** The development of the polar low (pointed with red arrows) between 18 UTC, 6 December 2013 and 12 UTC, 8 December 2013 through different analyses.

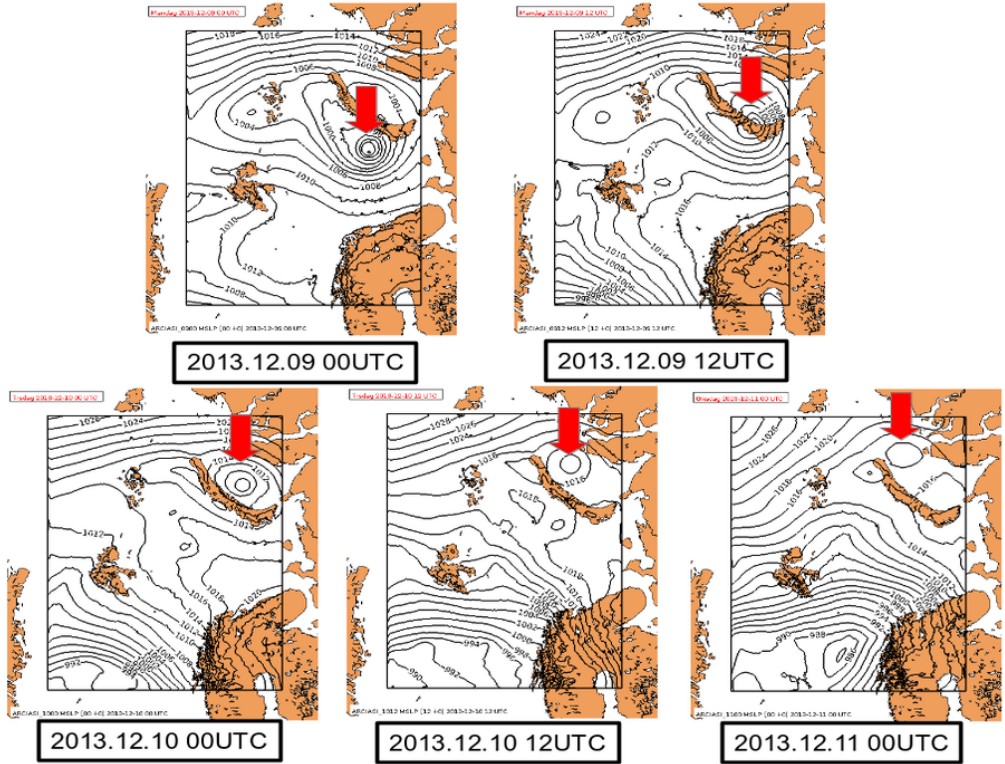

**Figure 18.** The development of the polar low (pointed with red arrows) between 00 UTC, 9 December 2013 and 00 UTC, 11 December 2013 through different analyses.

As seen in Figure 19, all runs predicted the centre of low pressure at slightly different positions and with slightly different intensities at the study time.

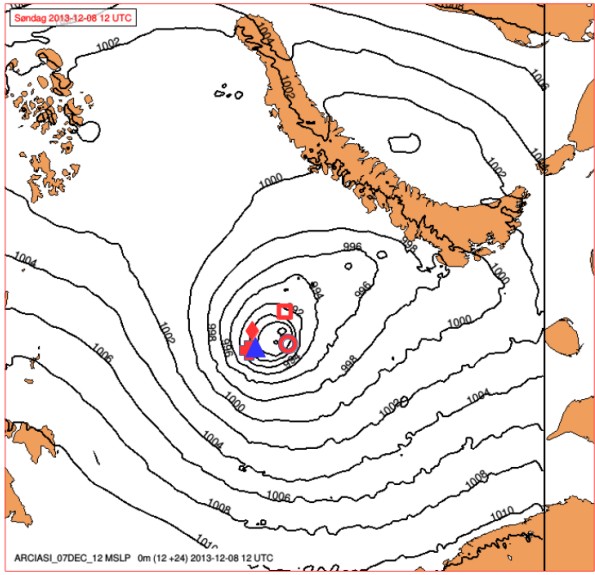

**Figure 19.** The position of the lowest points (shown next to the experiment names) of the forecasted low projected on the forecast using all observation (ARCIASI, 986 hPa). Inside the AROME-Arctic domain, Franz Josef land, Novaya Zemlya, Swalbard, and northern Norway can be seen. The different legends are as follows: not filled circle (ARCAMSUB, 986 hPa), not filled square (ARCATOVS, 990 hPa), plus-sign (ARCAMSUA, 984 hPa), blue triangle (ARCCONV, 984 hPa), and diamond (ARCREF, 986 hPa). Note that ARCIASI and ARCAMSUB predict very similar lows for both the position and intensity.

The position of the cross section was not fixed to best check each forecasted low. For simplicity, and accounting for the fact that the upper-tropospheric synoptic system is present in all forecasts, we zoom the lower tropospheric part in each forecast (Figure 20).

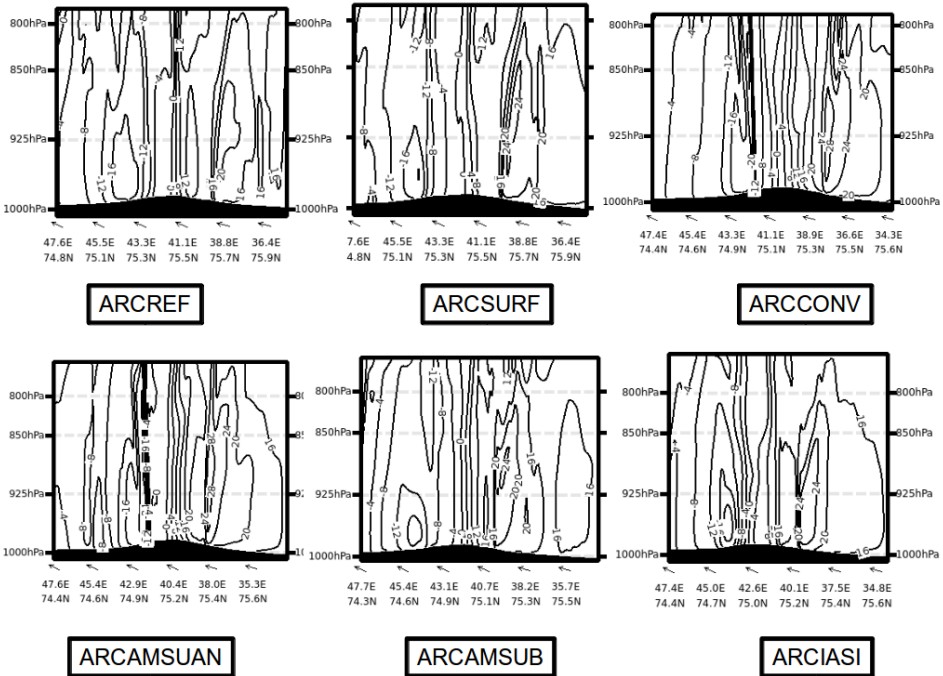

**Figure 20.** A vertical cross section of the 24-h forecasts valid at 12 UTC on 8 December 2013 issued from different experiments. The unit of wind is in m/s. The cross-sectional lines were chosen differently to better check the near surface vortice.

We can see that not all predicted lows prescribe well the near-surface vortex. It is not always clearly seen below 800 hPa. It is, of course, very subjective, and we think the near surface vortex is better seen in forecasts from ARCIASI, ARCAMSUB, and to some extent ARCREF but with different and lower intensities. The sensitivity of the forecast model to different observations is seen in the first case labelled "120612" in Figures 13–15, meaning the forecast from 6 December 12 UTC. The studied low passes through Q1, Q3, and Q4, while at a later stage, all quarters are somehow influenced. For example, we can see a large sensitivity on AMSUB and IASI in Q1 and Q3 and somehow on all quarters for the 6-, 12-, and 48-h forecasts.

## 4. Summary and Discussion

We observed that the H-A without data assimilation (surface and upper-air) has less accurate forecasts at surface and low tropospheric levels in verification against conventional observations. This deficiency may come from the fact that the surface schemes used in the coupling ECMWF model (e.g., Balsamo et al. [34]) and in the AROME-Arctic (Masson et al. [35]) model are different. Also, as discussed in Müller et al. [12] and Donier et al. [36], the applied land surface parameters defined by ECOCLIMAP 1 km (Masson et al. [37]) are part of the cause of a problematic model performance close to the surface and in the planetary boundary layers in an AROME and a SURFEX based models. See, for example, in Figure 4b, a considerable bias of 2-m temperature forecasts.

We used the operational ECMWF forecasts where more observations are used to build the initial state as lateral boundary conditions in this study. Despite of the relatively low resolution of these products, they have a relatively higher accuracy in upper-air than that of any shown experiments in this study. Therefore, we are talking about the relative impact of any checked observations in a

regional NWP. A more realistic impact could have been checked by using the LBCs with the same types of observations in each of the experiments. We (Roger Randriamampianina and Harald Schyberg) have been participating in an earlier study where the observing network impact was checked with the ECMWF-IFS experiments set up in the same way as for the regional models (but different models than used here, ALADIN and HIRLAM) over Europe. Part of the results of that study was published in Amstrup [38]. In such a study, the impact of each observing system would be expected to be clearer, and this turned out to be the case in this earlier study. Coordinated experiments with the ECMWF global model are being conducted, where we study the impact of observations in our model, applying such a strategy of identical observation scenarios. This will give a more complete overview of the impact of observing systems in our LAM (Limited-area model) through both data assimilation and the LBCs. Still, the present study allows us to draw some interesting conclusions. Without data assimilation, at least the surface one, it is not possible to produce accurate analyses/initial conditions and forecasts over the AROME-Arctic domain, especially near the surface and in the lower troposphere. Therefore, to improve the forecasting capability to protect lives and properties and to support activities in the Norwegian and Barents seas, data assimilation is needed to be part of the regional NWP.

Accounting for the fact that a regional model needs LBCs, which are products of global models using already a lot of observations, conventional observations are not sufficient to further improve the good performance obtained with surface assimilation only in our Arctic model setup. In our study, satellite radiance observations from microwave (ATOVS) and hyperspectral infrared (IASI) were added. Among the ATOVS instruments, the humidity sensitive AMSU-B/MHS showed a good impact, which was detected and demonstrated through different verifications, diagnostic tools, and case studies. Similarly to what was found in Randriamampianina et al. [9], IASI radiances provide a good impact in forecasting polar lows, which represent a forecast challenge thanks to the limited predictability connected to their small scales and vigorous convection and also to their impact due to abrupt change in local weather.

The MTEN study showed that, for stationary cases, the model is driven mainly by the LBC. As shown in the Section 2, in our regional data assimilation system, due to the relatively low model top, we use relatively less radiance observations and, so far, over cloud- and/or precipitation-free regions. Still, we can get a positive impact, especially in the case of complex meteorological events. The assimilation of cloudy radiances is now operational at ECMWF (Geer et al. [39]), and Meteo France is developing a two-step techniques (1-D-Bay + 4-D-Var) similar to the method which was implemented for radar reflectivity assimilation (Caumont et al. [40]; Wattrelot et al. [41]) to assimilate cloudy radiances but with a dedicated quality control and dedicated observation errors (Duruisseau et al. [42]). In the future, we plan to implement the ECMWF-IFS solution and expect that the impact of the microwave radiances will be larger.

To improve the composition of both microwave and high spectral infrared observations, we consider exploring the use of CrIS, ATMS, and microwave data from the FY-3 (FengYun-3) series of satellites. The latter appears to be very important for the 00 UTC run, where we have very few observations.

**Author Contributions:** Conceptualization, R.R.; methodology, R.R. and H.S.; validation, R.R. and M.M.; project administration, H.S.; writing—original draft, R.R.; writing—review and editing, R.R., M.M., and H.S.

**Funding:** This work was supported by the Arctic Climate Change, Economy, and Society research project ACCESS (contract No. 265863) part of the Ocean of Tomorrow 7th Framework Program of the European Union.

**Acknowledgments:** The computational resources were partly provided by ECMWF through its special project SPNORAND. The authors are thankful to Andrea Storto for improving the DFS plots and to Eoin Whelan in the the preparation of the verification against all observations. The Authors would like to thank the reviewers for their valuable help in improving the quality of the manuscript.

**Conflicts of Interest:** The authors declare no conflict of interest.

## Abbreviations

The following abbreviations are used in this manuscript:

| | |
|---|---|
| 3D-Var | Three-dimensional variational DA |
| ACARS | Aircraft Communications Addressing and Reporting System |
| ACCESS | Arctic Climate Change, Economy and Society project |
| AIREP | Aircraft Reports |
| ALADIN | Aire Limitée Adaptation dynamique Développement InterNational |
| AMDAR | Aircraft Meteorological Data Relay |
| AMSU-A | Advanced Microwave Sounding Unit-A |
| AMSU-B | Advanced Microwave Sounding Unit-B |
| APPLICATE | Advanced Prediction in Polar regions and beyond: modelling, observing system design and LInkages associated with a Changing Arctic climaTE |
| AROME | Application of Research to Operations at Mesoscale |
| ARCROSE | Arctic Research Collaboration for Radiosonde Observing System Experiment |
| ATMS | Advanced Technology Microwave Sounder |
| ATOVS | Advanced TIROS Operational Vertical Sounder |
| CrIS | Cross-track Infrared Sounder |
| DFS | Degree of Freedom for Signal |
| DRIBU | Drifting buoys |
| ECMWF | European Centre for Medium-range Weather Forecasting |
| ECOCLIMAP | Global database for soil-vegetation-atmosphere interactions |
| EUMETCAST | EUMETSAT primary dissemination system |
| EUMETSAT | European Organisation for the Exploitation of Meteorological Satellites |
| FOV | Field-of-view |
| GTS | Global Telecommunication System |
| HARMONIE | HIRLAM ALADIN Research on Mesoscale Operational NWP In Europe—convective-scale forecasting system |
| HIRLAM | High Resolution Local Area Modelling |
| IASI | Infrared Atmospheric Sounding Interferometer |
| IPY-THORPEX | International Polar Year –The Observing System. Research and Predictability Experiment |
| LBC | Lateral Boundary Conditions |
| METOP | Meteorological Operational Satellite |
| MetCoOp | Meteorological Co-operation on Operational NWP |
| MHS | Microwave Humidity Sounder |
| MTEN | Moist Total Energy Norm |
| NOAA | National Oceanic and Atmospheric Administration |
| NWP | Numerical Weather Prediction |
| OSE | Observing System Experiments |
| RTTOV | Radiative Transfer for TOVS |
| SURFEX | Surface Externalisée |
| SYNOP | Surface Synoptic Observations |
| TEMP | Upper-air soundings |

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
