# Peer review of "Observing System Experiments with an Arctic Mesoscale Numerical Weather Prediction Model"

_remotesensing, doi:10.3390/rs11080981_

Round 1

Reviewer 1 Report

This is a really useful paper analysing the impact of various observation
types on lam forecasts for the arctic region. It is original because it looks at
several methods to determine impact, both standard verification and other
methods e.g Energy norm approach.

I particularly like the fact that the cases include several varieties of
synoptic features.

I have a few suggestions and comments that i hope will help the authors improve the
paper.

1. in Data And Methods. I think it is important here to specify the source of the
ancilliary data - particularly Sea-Ice and Sea Surface Temperature fields. Are these analyses for example,
if so from where?

2. Table3c - assimilation of IASI channels. Can the authors add some detail on how the channel usage changes with cloud.
Also how is cloud determined in field of view

3. Graphs. Several of the graphs with forecast time do not include units of time e.g. 4(d), 5(b), 5(c), 6(b), 7. I think the units need to be
added. And also in Figure 20, please add units of wind.

4. Figure 5. One of the issues with impact studies in limited area regions is the acquisition of enough cases to present results that are statistically
significant. Cound the authors consider adding error bars to Figures 5(b) and (c). Then comment on the significance in the text.
It looks like the in 5b  for example that AMUSA/MHS together is statistically significantly better than just conventional observations, for the first few
hours at least.

5. Figure 19. The discussion on the polar low impacts is really useful. I suggest a lat/long or distance scale to be added on Figure 19.
It will aid those not familiar with this region.

6. Also on the Polar low discussion, it is interesting that the Polar low was captured by ArcRef with a realistic central pressure (Figure 19). Have the
authors checked that it existed in the ECMWF global fields used to force the model? It might demonstrate the usefulness in general of running LAMS in this region
to capture such important features.

7. In the conclusions you suggest in future work, what about considering the use of CrIS to augment the IASI data? it would improve coverage for small features like
polar lows for instance.

8. One final suggestion. The ref for RTTOV points to RTIASI-4. The standard reference for RTTOV i would suggest as an alternative is

Saunders, R., Hocking, J., Turner, E., Rayer, P., Rundle, D., Brunel, P., Vidot, J., Roquet,
P., Matricardi, M., Geer, A., Bormann, N., and Lupu, C., 2018:
An update on the RTTOV fast radiative transfer model (currently at version 12),
Geosci. Model Dev., 11, 2717-2737, https://doi.org/10.5194/gmd-11-2717-2018.

Author Response

We would like to thank the reviewer for the good comments and very useful suggestions. Below we give our answer to each comment/suggestion.

Reviewer 1:

This is a really useful paper analysing the impact of various observation types on lam forecasts for the arctic region. It is original because it looks at several methods to determine impact, both standard verification and other methods e.g Energy norm approach.

I particularly like the fact that the cases include several varieties of synoptic features.

I have a few suggestions and comments that i hope will help the authors improve the paper.

1. in Data And Methods. I think it is important here to specify the source of the ancilliary data - particularly Sea-Ice and Sea Surface Temperature fields. Are these analyses for example, if so from where?

We discuss the use of radiance observation over different surface conditions like for example over sea ice in this paper. The fields asked by the reviewer are both taken from the LBC, which is the IFS. We simulate the operational setting of coupling in this study, which means that the first LBC is a forecast. Note also that this report is about the performance of the early stage of our implementation. Later, we have worked out the state of the the sea ice in AROME-Arctic as described in Batrak and Müller (2018).

We didn’t do change in the text because these fields were not directly discussed in the Data and Methods.

2. Table3c - assimilation of IASI channels. Can the authors add some detail on how the channel usage changes with cloud.

We agree this information is missing in this paper. We think it’s meaningful to add explanation about this in the text, which we have done by adding the following sentence at the end of the sub-section 2.3:

For cloudy IASI pixels, active channels having a peak above the cloud top were assimilated. The cloud detection used in H-A is a version of McNally and Watts (2003) adapted to the IASI radiances as described in Collard and McNally (2009).

Also how is cloud determined in field of view

If the reviewer means the cloudiness information incorporated in the IASI BUFR file, it’s not use to define the cloudy IASI pixels in this study.

3. Graphs. Several of the graphs with forecast time do not include units of time e.g. 4(d), 5(b), 5(c), 6(b), 7. I think the units need to be added. And also in Figure 20, please add units of wind.

We corrected the figures 4(d), 5(b), 5(c), 6(b) and 7. In figure 20, the unit of wind is mentioned in the figure caption.

4. Figure 5. One of the issues with impact studies in limited area regions is the acquisition of enough cases to present results that are statistically significant. Cound the authors consider adding error bars to Figures 5(b) and (c). Then comment on the significance in the text.

Unfortunately, we don’t have the possibility to add significance test on these figures. We have separate significance test on the performance of AMSU-A, AMSU-B/MHS, AMSU-A+AMSU-B/MHS and IASI as shown below. The significance test is shown according to the performed experiments. AMSU-A, AMSU-B/MHS and ATOVS on top of conventional data. IASI on top of conventional + ATOVS. As we can see the differences are not significant, except for the full use of radiances, where we can see an almost significant difference at 12 hour forecast (ARCIASI vs ARCATOVN).

We added in the text, after line 264, the following sentence:

The difference in dew point temperature at 700 hPa between these runs is not significant.

When checking the plots in the attachment please pay attention to the legends. We hope it’s clear.

It looks like the in 5b  for example that AMUSA/MHS together is statistically significantly better than just conventional observations, for the first few hours at least.

Figure 5 (b) and 5 (c) is showing the analysed and forecast values, and not the differences. So, as shown in the figures above the differences are not significant. But, in the way we show the results, we see the respective change in the atmospheric state with respect to the added observations.

5. Figure 19. The discussion on the polar low impacts is really useful. I suggest a lat/long or distance scale to be added on Figure 19. 
It will aid those not familiar with this region.

We recreated this figure with larger area, where Novaya Zemlya is well seen, as shown in the attachment. The names of the Arctic islands inside H-A domain are now mentioned in the figure caption as well. We hope the readers can orientate themselves in this figure and this is fine with the reviewer. We think we need to zoom to better show what we want.

6. Also on the Polar low discussion, it is interesting that the Polar low was captured by ArcRef with a realistic central pressure (Figure 19). Have the authors checked that it existed in the ECMWF global fields used to force the model? It might demonstrate the usefulness in general of running LAMS in this region to capture such important features.

Unfortunately, we haven’t checked the performance of the IFS in this case. But, we learned from our colleagues at forecasting division that IFS is more efficient in medium range (from day 3) and less in short range (up to day 2). Taking this into account the above opinion and keeping in mind that we simulate our operational use of the LBCs, 24 hour forecast is coupled with, the earliest, 30-hour forecast. We expect that IFS is quite reliable at this range.

7. In the conclusions you suggest in future work, what about considering the use of CrIS to augment the IASI data? it would improve coverage for small features like polar lows for instance.

We agree, we added data from CrIS, ATMS and microwave from FY-3 (FengYun-3) series satellites to be explored in the future.

8. One final suggestion. The ref for RTTOV points to RTIASI-4. The standard reference for RTTOV i would suggest as an alternative is

Indeed thanks for this. We changed the reference.

Saunders, R., Hocking, J., Turner, E., Rayer, P., Rundle, D., Brunel, P., Vidot, J., Roquet, 
P., Matricardi, M., Geer, A., Bormann, N., and Lupu, C., 2018: 
An update on the RTTOV fast radiative transfer model (currently at version 12), 
Geosci. Model Dev., 11, 2717-2737, https://doi.org/10.5194/gmd-11-2717-2018.

References:

Batrak, Y., and Müller, M. (2018). Atmospheric response to kilometer‐scale changes in sea ice concentration within the marginal ice zone. Geophysical Research Letters, 45. https://doi.org/10.1029/2018GL078295.

Collard, A., McNally, A.: The assimilation of infrared atmospheric sounding interferometer radiances at ECMWF.  Q. J. R. Meteorol. Soc., 2009 , 135 , 1044–1058, doi: 10.1002/qj.410.

McNally, A., Watts, P.: A cloud detection algorithm for high-spectral-resolution infrared sounders. Q. J. R. Meteorol. Soc., 2003, 129, 3411–3423, doi: 10.1256/qj.02.208.

Reviewer 2 Report

Dear colleagues,

I accepted to review the paper “Observing system experiments with an Arctic mesoscale numerical weather prediction model” with the clear consciousness of not being myself expert in data assimilation. The topic is very significant for NWP and related papers are of a good profit as well for  scientific community, as for NWP community. To generalise briefly, the paper is focused on the impact of different observations on the analyses and forecasts of the limited area AROME-Arctic model. It consists of a computation of degrees of freedom for signals on the analysis, an utilisation of an energy norm-based approach applied to the forecasts, a verification against observations and case studies. The general conclusion of the study is the need of data assimilation to be part of the regional NWP, and that the conventional observations are not sufficient for the further improvement of model performance. Good impact of satellite radiance observations from microwave and hyper spectral infrared on specific forecast over the AROME-Arctice domain are demonstrated.

In my opinion the paper is written in a well grounded scientific manner and should be accepted for publication. I  did not find any mistakes, except the typo in page 3 line 91 where “except” should be in the place of “expect”.

I have one question. In my understanding, in figures 4 and 5 the mean RMSE, STDV and BIAS for the 2m temperature and dew point for the four daily model runs (at 00, 06, 12 and 18 UTC) are shown. What are the results for each model run separately? Usually, there is a diurnal trend of model performance in forecasting the temperature and humidity at 2m, which could be masked when different model runs are averaged.

p { margin-bottom: 0.1in; line-height: 120%; }

p { margin-bottom: 0.1in; line-height: 120%; }

Author Response

@page { margin: 0.79in } p { margin-bottom: 0.1in; line-height: 120% }

We would like to thank the reviewer for the good comments and very useful suggestion. Below we give our answer to each comment/suggestion

Reviewer 2:

Dear colleagues,

I accepted to review the paper “Observing system experiments with an Arctic mesoscale numerical weather prediction model” with the clear consciousness of not being myself expert in data assimilation. The topic is very significant for NWP and related papers are of a good profit as well for  scientific community, as for NWP community. To generalise briefly, the paper is focused on the impact of different observations on the analyses and forecasts of the limited area AROME-Arctic model. It consists of a computation of degrees of freedom for signals on the analysis, an utilisation of an energy norm-based approach applied to the forecasts, a verification against observations and case studies. The general conclusion of the study is the need of data assimilation to be part of the regional NWP, and that the conventional observations are not sufficient for the further improvement of model performance. Good impact of satellite radiance observations from microwave and hyper spectral infrared on specific forecast over the AROME-Arctice domain are demonstrated.

In my opinion the paper is written in a well grounded scientific manner and should be accepted for publication. I  did not find any mistakes, except the typo in page 3 line 91 where “except” should be in the place of “expect”.

Thank you, we corrected this.

I have one question. In my understanding, in figures 4 and 5 the mean RMSE, STDV and BIAS for the 2m temperature and dew point for the four daily model runs (at 00, 06, 12 and 18 UTC) are shown. What are the results for each model run separately? Usually, there is a diurnal trend of model performance in forecasting the temperature and humidity at 2m, which could be masked when different model runs are averaged.

Attached, we show the Fig 5 (c) at 00 and 12 UTC. One can see that larger impact of IASI is observed at 00 UTC. Since we don’t have the IASI data at 00 UTC, we see the impact through the first-guess from 21 UTC (see Table 2). Smaller impact at 12 UTC, probably because many satellites are available at this assimilation time.

Regarding the T2m score, the model performance is very similar for both 00 and 12 UTC runs. Similarly, the model scores on humidity 2m are similar from 00 and 12 UTC (not shown).
